# The Function of Flavonoids in the Diurnal Rhythm under Rapidly Changing UV Conditions—A Model Study on Okra

**DOI:** 10.3390/plants10112268

**Published:** 2021-10-22

**Authors:** Susanne Neugart, Mark A. Tobler, Paul W. Barnes

**Affiliations:** 1Division Quality and Sensory of Plant Products, Georg-August-Universität Göttingen, 37075 Goettingen, Germany; 2Department of Biological Sciences, Loyola University New Orleans, 6363 St. Charles Avenue, New Orleans, LA 70118, USA; tobler@loyno.edu (M.A.T.); pwbarnes@loyno.edu (P.W.B.)

**Keywords:** Dualex, diurnal rhythm, UV exclusion, quercetin-3-xylosyl-glucoside

## Abstract

Flavonoids are favored compounds in plant responses to UV exposure and act in UV absorption and antioxidant activity. Here, it was investigated, with okra as a model species, how fast plants can react to changing UV conditions and to what extent these reactions take place. Okra (*Abelmoschus esculentus*) plants were exposed to either full or nearly no UV radiation. The diurnal rhythm of the plants was driven by the UV radiation and showed up to a 50% increase of the flavonoid content (measured optically in the +UV plants). This was reflected only in the trends in UV-absorption and antioxidant activity of the extracts but not in the soluble flavonoid glycosides and hydroxycinnamic acid derivatives. In a second experiment, a transfer from a −UV to a +UV condition at 9:00 CDT showed the immediate start of the diurnal rhythm, while this did not occur if the transfer occurred later in the day; these plants only started a diurnal rhythm the following day. After an adaptation period of seven days, clear differences between the +UV and -UV plants could be found in all parameters, whereas plants transferred to the opposite UV condition settle between the +UV and -UV plants in all parameters. Broadly, it can be seen that the flavonoid contents and associated functions in the plant are subject to considerable changes within one day and within several days due to the UV conditions and that this can have a considerable impact on the quality of plant foods.

## 1. Introduction

UV radiation is an important regulator of adaptation processes in plants. Plants are often very well adapted to the UV conditions of their regions of origin [1]. Climate change will lead to changes in the amount and variability of solar UV radiation that these plants receive. For instance, studies show that due to changes in cloud formation, higher UV radiation intensities are more likely to occur in mid-latitude regions in the coming years, accompanied by shading due to the formed clouds [2]. One of the best-known plant responses to increased UV radiation is the formation of flavonoids and other phenolic compounds [3].

Flavonoids and hydroxycinnamic acids occur ubiquitously in plants and are significant in plant-environment interactions. Yet, not all compounds are affected to the same degree by UV as they exhibit variations in specific characteristics, which are also linked to functions such as UV-absorbance and antioxidant activity. For example, quercetin glycosides show increased antioxidant activity compared to kaempferol glycosides [4,5], but these compounds may not necessarily differ in UV-absorbance properties [4]. Other characteristics may relate to the acylation pattern or glycosylation pattern of flavonoid glycosides [6,7,8] but may also depend on the duration of UV-B treatment and the acclimation time of the plant [6,9]. Glycosylation decreases the antioxidant activity of flavonoids [4,10]. In addition, the response of flavonoid glycosides depends on the type of phenolic acid with which the flavonol glycoside is acylated. For example, in pak choi, the total flavonoid content increases with UV-B exposure, but kaempferol glycosides acylated with ferulic acid, hydroxyferulic acid, or sinapic acid did not respond to UV exposure at 22 °C [7]. Phenolic compounds containing a catechol structure also exhibit particularly high antioxidant activities [6,11,12]. However, in *Brassica* species, the concentration of kaempferol-3-*O*-sinapoyl sophoroside-7-*O*-glucoside is not affected in UV-exposed plants [8,13,14], while other quercetin glycosides and kaempferol glycosides are increased by UV radiation [8,14]. Quercetin and kaempferol are the main dietary flavonoids and are of special interest due to their antioxidant activity, as well as anti-inflammatory and anticancerogenic effects in humans [15,16,17].

Okra, *Abelmoschus esculentus* (L.) Moench, belongs to the family of Malvaceae. This cultivated plant is of African origin but is now grown in different tropical and warm temperature regions of the world, like Greece, Turkey, Iran, Egypt, India, Japan, Philippines, and southern United States [18]. Of the more than 30 species studied to date, okra has been shown to exhibit the greatest changes in epidermal UV-A transmittance during the day, and therefore this species serves as a model to study the mechanisms and kinetics of rapid UV protective responses. This diurnal rhythm appears to be mainly driven by UV radiation [19,20]. These studies have shown that the diurnal decrease in UV-A transmittance (from dawn to midday) is associated with an increase in soluble UV-absorbing compounds throughout the leaf as well as specific quercetin glycosides [21]. A number of monoglycosylated quercetin glycosides and sinapic acid derivatives have been detected in okra [21,22]. It is, therefore, a suitable model plant as quercetin glycosides are the main dietary flavonoids in humans. In a previous experiment, a 50% increase of flavonoid content was detected within a few hours in the diurnal rhythm of okra, which is enormously rapid [23]. Flavonoid contents show a negative correlation to UV-A transmittance [23], suggesting that flavonoids act as UV-absorbing compounds [4,24]. However, flavonoids also act as antioxidants in plants [12] and humans [15,16,17], and okra is a model plant due to its flavonoid profile and ability to respond to changing UV conditions.

The present study aims to clarify whether rapid changes in flavonoid contents measured by optical methods are related to (1) soluble flavonoid glycosides and hydroxycinnamic acids, (2) UV-absorption, and (3) antioxidant activity. In addition, it will be clarified how fast the diurnal rhythm of plants can be modified by changing the UV conditions and whether it plays a role in which state of the diurnal rhythm the plant is in.

## 2. Results

Two experiments were set up. In Experiment 1, the diurnal rhythm of okra plants grown under +UV conditions and -UV conditions were studied, while in Experiment 2, plants were adapted to a UV condition and transferred to the opposite UV condition to follow the changes triggered (Appendix A).

### 2.1. Morphological Parameters

Overall, most of the measured morphological parameters showed significantly or tendentially lower values in +UV plants compared to -UV plants (Table 1). Plants transferred to the other respective UV condition ranged between the +UV and -UV plants for the midday values in Experiment 2, where the changes due to a transfer of plants to the opposite UV condition were investigated. It is of note that in the values of the 9:00 CDT transferred and harvested plants, the absolute lowest values for plant height, leaf area, and fresh and dry mass were found in the -UV/+UV plants.

### 2.2. Flavonoids

Diurnal changes were detected in the youngest mature leaf in Experiment 1 (Figure 1A). The flavonoid content was increased to approximately 1.5-fold, having its highest absolute value at 15:00 CDT. Such an effect could not be found for the -UV plants. Furthermore, in Experiment 2, the changes due to a transfer of plants to the opposite UV condition were investigated. It could be shown that +UV plants follow the diurnal rhythm over three days, while -UV plants did this only to a very small extent (Figure 1B). It is interesting that a transfer of plants from -UV conditions to +UV conditions at 9:00 CDT led to an uptake of this diurnal rhythm within 2 h. The flavonoid content was successively increased, but after three days, it did not come close to the concentrations of the +UV plants that stayed in their UV condition. In contrast, plants transferred from +UV conditions to -UV conditions showed a flattening of the diurnal rhythm within three days, with no significant decrease in flavonoid content. A transfer of the plants at 13:00 CDT showed a similar picture, with the exception that on day one, the plants showed equal flavonoid contents with the previous UV treatment (Figure 1B). They started a diurnal rhythm on the following day. In general, the -UV plants showed the lowest flavonoid contents, and the +UV plants showed the highest flavonoid contents (Figure 1C). After seven days of acclimation, the transferred plants had flavonoid concentrations in intermediate ranges. In addition, the -UV/+UV showed significantly higher trends in flavonoid concentrations at 9:00 CDT and 13:00 CDT, compared to the +UV/-UV plants. Furthermore, the corresponding quercetin glycosides and kaempferol glycosides (e.g., the main quercetin-3-*O*-glucoside-xyloside compound in okra) also had high concentrations. The concentrations of soluble flavonoid glycosides and hydroxycinnamic acid derivatives behaved differently from the optically measured flavonoid contents. In Experiment 1 (investigating the diurnal rhythm), increasing concentrations of flavonoid glycosides and hydroxycinnamic acid derivatives were found at 11:00 CDT, followed by a sharp decrease at 13:00 CDT and 15:00 CDT (Table 2). Towards 17:00 CDT in the evening, the concentrations increased again. Here, the +UV plants showed a stronger expression on the effect than the -UV plants. In Experiment 2, targeting changes in diurnal rhythm due to a transfer to the opposite UV condition, the data were significantly affected by the time of harvest. While in the morning (9:00 CDT), the concentrations were often slightly lower than in the afternoon (13:00 CDT) in plants under +UV conditions; however, the opposite was found for plants under -UV conditions (Table 3). In only a few compounds, such as isorhamnetin-3-*O*-glycoside and sinapoyl-glucoside, +UV plants were higher than -UV plants at 9:00 CDT. At 13:00 CDT, the concentrations of most flavonoid glycosides were higher in the +UV plants than in the -UV plants. The plants in which the UV conditions were changed had medium concentrations of flavonoid glycosides at midday, with the -UV/+UV variant having higher concentrations of both. The hydroxyferuloyl glucoside was not affected.

### 2.3. UV-Absorption

There was a higher UV absorption at 330 nm than at 370 nm. Overall, in Experiment 1, targeting the diurnal rhythm, the +UV plants showed a diurnal rhythm at 330 nm, while no diurnal rhythm was found in the -UV plants. No diurnal rhythm was found at 375 nm either, despite the slight changes (Figure 2A). A higher UV absorption of +UV plants compared to -UV plants was also found in Experiment 2, targeting changes in the diurnal rhythm due to a transfer to the opposite UV condition, for 330 nm and 375 nm at both harvest times (Figure 2B). The plants in which the UV conditions were reversed for seven days are between +UV plants and -UV plants. The diurnal effects were smaller in Experiment 2 than in Experiment 1.

### 2.4. Antioxidant Activity

Overall, the results of three different antioxidant assays for Experiment 1, investigating the diurnal rhythm, show higher antioxidant activity for +UV plants than for -UV plants (Figure 3A). For TEAC and FRAP, a slightly different diurnal rhythm was shown for both +UV plants and -UV plants. The DPPH assay, on the other hand, showed a decrease in antioxidant activity over the diurnal cycle. Higher antioxidant activity was also observed in +UV plants compared to -UV plants in Experiment 2, targeting changes in diurnal rhythm due to a transfer to the opposite UV condition in all assays (Figure 3B). The plants that were transferred to the other UV condition also showed medium antioxidant activity. It can be seen that the extent of the diurnal rhythm was lower in Experiment 2 than in Experiment 1.

### 2.5. Relation of Flavonoids and Function in the Diurnal Rhythm

In the +UV plants, UV absorbance, TEAC, and FRAP showed strong correlations (Table 4). This was not the case for DPPH. There were some weak correlations between UV absorption and specific flavonoid glycosides and hydroxycinnamic acid derivatives, whereas there was no correlation between antioxidant activity and the specific flavonoid glycosides and hydroxycinnamic acid derivatives. However, the flavonoid glycosides and hydroxycinnamic acids showed strong correlations with each other but not with the flavonoid content measured optically. Comparable results were also found for the -UV plants, but to a lesser extent (Table 5).

## 3. Discussion

### 3.1. Morphological Parameters

Plants show a variety of responses to UV radiation. For example, morphological parameters, such as leaf area or plant height and fresh mass, are minimized by UV radiation in both experiments presented. In soybean, for example, reductions in leaf mass and plant height were found due to UV [25]. This phenomenon has led some to consider UV radiation a stressor [26], but these morphological changes often occur without changes in shoot biomass accumulation or photosynthesis, suggesting a non-damaging mechanism underlying these responses.

### 3.2. Flavonoids

It has been shown previously that flavonoids can be increased by moderate UV radiation [27,28,29]. The flavonoid content, which is measured optically, refers to the epidermal cell wall-bound and soluble flavonoid glycosides and hydroxycinnamic acid derivatives. In our results, flavonoid content increased in a diurnal rhythm up to 15:00 CDT and, matching this, absorbance at 330 nm and antioxidant activity in the TEAC and FRAP assays also increased. Such diurnal rhythms, with respect to UV absorbing compounds, have already been shown in okra and tomato, as well as other plants [19,30,31]. Many, but not all, plants have also shown the ability to quickly adjust UV screening in response to short-term fluctuations in solar UV radiation [31,32]. These rapid adjustments in leaf optical properties are fully reversible [21]. Soluble flavonoid glycosides and hydroxycinnamic acid derivatives, measured from whole leaf extracts, did not fit into this picture of diurnal rhythm in Experiment 1. In okra, quercetin glycosides and kaempferol glycosides are present, but the concentration of hydroxycinnamic acid derivatives is higher than that of flavonols [22]. Quercetin-3-xylosyl-glucoside has previously been identified in okra by NMR [33]. It is suggested that in Experiment 1, the UV radiation was very high at 13:00 CDT and the following hours, and thus, the flavonoids formed were directly consumed as antioxidants and were self-oxidized. In other studies, there was often agreement of Dualex values with, e.g., total phenolic content or flavonoid groups [34,35], and higher UV radiation was also associated with higher flavonoid content [36]. We cannot support this unconditionally. Even more precisely, Agati et al. [37] showed, in wine, that there is a correlation of flavonoid contents measured with the Dualex and soluble flavonoids but not with the soluble hydroxycinnamic acids. Consequently, it can be shown in this study that the soluble flavonoid glycosides and hydroxycinnamic acid derivatives (1) only account for a part of the UV response, and the UV absorption and antioxidant activity could also be taken over by other compounds or cell wall-bound phenols or (2) are oxidized and, thus, analytically not detectable with the used method. This results in the few correlations of specific flavonoid glycosides and hydroxycinnamic acid derivatives and functions, such as UV absorption and antioxidant activity, and flavonoid content, measured optically in Experiment 1. Other studies, nevertheless, found good agreement on the latter [35,38], but there are also studies that found no agreement [39]. However, in Experiment 2, the soluble flavonoid glycosides and hydroxycinnamic acid derivatives showed a clear influence of UV radiation due to increased flavonoid glycosides and hydroxycinnamic acid derivatives at 13:00 CDT, while the morning values were much more diffuse, supporting the theory that the flavonoid glycosides and hydroxycinnamic acid derivatives are synthesized on a daily basis according to demand. In Experiment 2, the UV-B values were lower (3.0 kJ m^−2^ d) compared to Experiment 1 (4.5 kJ m^−2^ d). It is not yet clear if there is a threshold, e.g., for the oxidation or delocalization of flavonoid glycosides. Nevertheless, the response to altered UV conditions can occur within 2 h, in the 9:00 CDT transfer of plants, whereas plants transferred at midday did not begin an altered diurnal rhythm until the next day. Overall, the -UV/+UV transferred plants did not show the same flavonoid contents, UV absorptions, and antioxidant activities even after seven days, as plants that were exposed to UV radiation all the time. That adaptation to elevated UV can take several days and has already been shown for grapevine, bell pepper, and fava bean [40,41,42,43,44]. These findings indicate that the response to UV in okra is a highly plastic and dynamic trait that can change over multiple time scales in response to variation in solar UV radiation. Flavonoids are often increased, whereas hydroxycinnamic acid has a weaker reaction in comparison. It can be concluded that light dose and flavonoid content have a dose-response relationship [45,46,47,48]. Furthermore, the type of UV also plays a role. Both UV-A and UV-B are able to increase flavonoid levels, but UV-B has stronger efficacy [49]. However, there are also other factors influencing flavonoid content, for example, negative correlations with nitrogen concentrations and pathogen infestation [50,51,52,53]. This highlights the regulatory effect that UV radiation has on plant mechanisms as presented by flavonoid glycosides and hydroxycinnamic acids here. A longer-acting moderate UV radiation can have a positive effect on the plant’s constituents, whereas high UV might be a stressor [54].

### 3.3. UV-Absorption

In rye, it was shown that epidermal flavonoid contents have a strong correlation to UV-A and UV-B absorption, but not mesophyll flavonoids [55,56]; likewise, this was found in sea buckthorn at 300 nm [57]. Meanwhile, wine flavonols have a better correlation to absorbance at 314 nm and 360 nm than hydroxycinnamic acid derivatives [27]. This can also be confirmed for okra, even if the correlations are rather weak. Overall, higher UV absorptions are found at 305 nm or 330 nm instead of 375 nm in many species [25,31,47,58], which can be explained in okra by the high concentration of hydroxycinnamic acid derivatives on the one hand and by the accumulation of absorption maxima of the different other compounds at 330 nm on the other hand. The increase in UV absorption at higher UV (e.g., +UV plants in Experiment 2) resulted in a lower UV transmittance (Day 1994), as was also shown in okra [19,23,30,31]. The rapid adjustments of flavonoid content measured optically are only partly reflected by the UV-absorption of the leaf extracts, underlining the optical effect of epidermal cell-wall bound phenolics in UV responses not measured here.

### 3.4. Antioxidant Activity

It is known that UV radiation can increase flavonoids in different species and simultaneously their antioxidant activity [59]. Compared to the TEAC assay, hydroxycinnamic acids and kaempferol glycosides are underrepresented in DPPH, while quercetin glycosides are more represented [10]. Since okra has predominantly quercetin glycosides instead of kaempferol glycosides, the quantitative difference between TEAC and DPPH can be explained. For example, the strong antioxidant activity of okra flower extract was shown in a DPPH assay [60], which can be confirmed by our results. Comparable results were found in a study on reference standards for DPPH and FRAP [10], but the results of the FRAP assay for okra are significantly lower compared to DPPH. The different reference substances used in both studies could be an explanation for this phenomenon. While +UV and -UV plants could be clearly distinguished, there was little effect of diurnal rhythm, suggesting that (1) there are other compounds that contribute to antioxidant activity, such as vitamin C, or (2) there are concurrent oxidation processes that interfere with the results.

### 3.5. Relation of Flavonoids and Function in the Diurnal Rhythm

Although there are no significant correlations between flavonoid content and UV absorbance at 330 nm or antioxidant activity in the TEAC assay, there is a tendential relationship, especially in +UV plants. Such diurnal rhythms, with respect to UV absorbing compounds, have been shown in okra, tomato, and other plants [19,30,31]. Both UV absorption and antioxidant activity show a relationship with flavonoid adaptation to the altered UV conditions [59]. This reflects the correlations of UV-absorption and antioxidant activity in the +UV plants in particular. Furthermore, this is also visible in Experiment 2, even in the two transferred plant sets. Soluble flavonoid glycosides and hydroxycinnamic acids contributed to the UV-absorbance and antioxidant activity, which is in line with results from linden trees where the antioxidant activity was the main function of flavonoids [4]. In the present experiment, we found that PAR (in the -UV treatment) plays a role in the formation of flavonoids and other phenolics [61], but to a much lesser extent in the UV radiation (in the +UV treatments). Furthermore, the higher concentrations of hydroxycinnamic acid derivatives in okra could lead to higher UV absorption, especially in the range of about 330 nm of the plant in general [62].

## 4. Materials and Methods

### 4.1. Plant Experiments

Studies were conducted on okra (*Abelmoschus esculentus*) cv. Clemson Spineless #80. The plants were germinated in potting soil in a roof-top greenhouse at ambient conditions. After germination, the plants were grown further in the greenhouse for another 7 days. The plants were then transferred outdoors on day 7 for the UV-light experiments in New Orleans, LA, USA (29°57′3.8340″ N, 90°4′17.5188″ W). Different solar radiation treatments were achieved by frames that held various types of plastic film in nearly full UV (+UV) conditions by a UV-transparent film (Aclar type 22 A, 0.038 mm thick, Honeywell, Pottsville, PA, USA) or in UV-excluding (-UV) conditions by a UV-blocking film with a cut off near 390 nm (CFC, 0.051 mm thick, LK Technologies, Maple Heights, OH, USA). For further information on the films, please see Appendix A. The plants were watered daily, if needed, and fertilized once a week with an all-purpose liquid fertilizer (3-1-2; N-P-K) throughout their growing time. The climate conditions, namely UV, photosynthetically active radiation (PAR), and temperature, are given in Appendix A.

#### 4.1.1. Experiment 1: Diurnal Rhythm

After 4 weeks, diurnal measurements of flavonoid content (Dualex) were performed on the adaxial (upper) surfaces of the youngest mature, healthy leaves of 10 individual plants (9:00, 11:00, 13:00, 15:00, and 18:00 CDT). Furthermore, the same leaves were harvested, and soluble flavonoids and hydroxycinnamic acids (HPLC), UV-absorbance, and antioxidant activity of leaf extracts (both spectrophotometer) were measured from methanolic extracts of these leaves (Appendix A). In addition, morphological parameters such as plant height, internode length, leaf area, fresh mass, and dry mass of 10 individual plants from the 13:00 CDT harvest were measured.

#### 4.1.2. Experiment 2: Changes in Diurnal Rhythm Due to Transfer to the Opposite UV Condition

After three weeks, plants were either left in their original UV condition or transferred to the other UV treatment. This created four categories of UV exposure: +UV, +UV/-UV, -UV/+UV, -UV (Appendix A). To determine if diurnal rhythm affected the results, this transfer was done either at 9:00 CDT in the morning or 13:00 CDT (solar noon). Flavonoid content (Dualex) was monitored for three days at the exact same location on the youngest mature, healthy leaves of 10 individual plants. After a total of 7 days, the plants were harvested either at 9:00 CDT in the morning or at 13:00 CDT at solar noon. All parameters were determined as in Experiment 1.

### 4.2. Morphological Parameters

Plant height and internode length were measured with a folding rule for the 10 plants. Leaf area was recorded by digital photographs and a later determination via Fiji image analysis. The fresh and dry masses were recorded for the youngest fully developed leaves after the removal of a leaf disc of 0.75 cm^2^. In addition, the dry mass was measured for this leaf disc after extraction for the study of UV-absorption and antioxidant activity.

### 4.3. Flavonoid Content

Non-destructive measurements of flavonoid content were performed using a Dualex fluorimeter (Dualex Scientific+; FORCE-A, Orsay, France). In Experiment 1, ten new plants were taken for harvest at each time point. In Experiment 2, after transferring to the different UV conditions, we examined the exact same location of the leaf every hour from 9:00 to 17:00 for three days and during harvest on day 7.

### 4.4. Extraction and Chemical Analysis of Flavonoid Glycosides and Hydroxycinnamic Acid Derivatives

To analyze the phenolic compounds, 20 mg of a lyophilized okra sample was extracted with 60% methanol (per analysis, Carl Roth GmbH, Karlsruhe, Germany) according to Neugart et al. [63]. For the quantitative analysis of the flavonoid glycosides and hydroxycinnamic acid derivatives, an HPLC series 1100 (Agilent Technologies, Waldbronn, Germany) was used, according to Neugart et al. [64]. The tentative identification of flavonoid glycosides and hydroxycinnamic acid derivatives was performed using an amazon SL ion trap mass spectrometer (Bruker Daltonics GmbH & Co. KG, Bremen, Germany) in negative ionization mode, according to Neugart et al. [64]. Standards (chlorogenic acid, quercertin 3-glucoside, and kaempferol 3-glucoside; Roth, Karlsruhe, Germany) were used for external calibration curves. Results are presented as mg g^−1^ dry weight (DW).

### 4.5. UV-Absorbance

Fresh leaf discs (0.75 cm^2^) were extracted with 5 mL cold acidified methanol. Spectrophotometric determination of total UV absorption (A280-400 nm) was conducted for the extracts of 10 leaves per treatment and time point. The wavelength 330 nm was linked to hydroxycinnamic acid derivatives, while the wavelength 375 nm was linked to the flavonoid glycosides. All absorption measurements were carried out using a UV/Vis spectrophotometer (Model DU640; Beckman Coulter, Inc., Fullerton, CA, USA).

### 4.6. Antioxidant Activity

Fresh leaf discs (0.75 cm^2^) were extracted with 5 mL cold acidified methanol.

#### 4.6.1. The TEAC Assay (Trolox Euivalent Antioxidant Capacity) was Performed with the Following Method:

An ABTS-working solution (0.06 mM) with 9.6 mg ABTS, 1.66 mg potassium persulfate, 25 mL milliQ water, and 20 mL methanol was prepared. Each 100 µL of diluted plant extracts (1:10) were transferred to a 4 mL cuvette and mixed with 2400 µL ABTS-working solution. After 6 min, the extinction was measured at 734 nm with a UV/Vis spectrophotometer (Model DU640; Beckman Coulter, Inc., Fullerton, CA, USA). Results were evaluated using a Trolox^®^ (Sigma-Aldrich, Inc., St. Louis, MO, USA) calibration curve (2.5 mM) and expressed as mM Troloxequivalent g-1 dry matter.

#### 4.6.2. The FRAP Assay (Ferric Reducing Antioxidant Power) was Performed with the Following Method:

The FRAP working solution was prepared by mixing 250 mL of acetate buffer (300 mM, pH 3.6), 25 mL TPTZ solution (10 mM TPTZ in 40 mM HCl), and 25 mL of FeCl3 (20 mM in water solution). Each 100 µL of diluted plant extracts (1:10) were transferred to a 4 mL cuvette and mixed with a 2400 µL FRAP-working solution. After 30 min, the extinction was measured at 593 nm with a UV/Vis spectrophotometer (Model DU640; Beckman Coulter, Inc., Fullerton, CA, USA). Results were evaluated via the Trolox^®^ calibration curve (2.5 mM) and expressed as mM Troloxequivalent g-1 dry matter.

#### 4.6.3. The DPPH Assay (2,2-Diphenyl-1-picrylhydrazyl Assay) Was Performed with the Following Method:

A DPPH working solution was prepared with 5.91 mg DPPH and 25 mL methanol. Each 100 µL of diluted plant extracts were transferred to a 4 mL cuvette and mixed with a 2400 µL DPPH working solution. After 30 min, the extinction was measured at 515 nm with a UV/Vis spectrophotometer (Model DU640; Beckman Coulter, Inc., Fullerton, CA, USA). Results were evaluated via the Trolox^®^ calibration curve (2.5 mM) and expressed as mM Troloxequivalent g-1 dry matter.

### 4.7. Statistical Analysis

All measurements were done on ten independent biological replicates with three independent technical replicates in the lab. All data included were statistically analyzed by a one-factorial ANOVA using Tukey’s HSD test at a significance level of 5%. Pearson correlation coefficients (r) were used to examine the relationships between UV-absorption, antioxidant activity, specific flavonoid glycosides, hydroxycinnamic acid derivatives, and flavonoid content of okra at a significance level of 5%. Calculations were performed using StatisticaTM for WindowsTM (version 9.0, Statsoft Inc., Tulsa, Okla).

## 5. Conclusions

This mechanistic study on okra shows that UV radiation has a significant effect on the diurnal rhythm of plants, but plants adapt relatively quickly to safely changing UV conditions. In okra, the optically measured flavonoid contents can show strong differences in diurnal rhythms. The UV absorption, antioxidant activity, soluble flavonoid glycosides, and hydroxycinnamic acid derivatives of leaf extracts do not necessarily show the same pattern, indicating the involvement of cell wall-bound phenols or oxidation processes. Large differences between +UV and -UV can be detected reliably, and distinctions can be found in the optically measured flavonoid contents, soluble flavonoid glycosides, hydroxycinnamic acid derivatives, UV absorption, and antioxidant activity. It was shown that the diurnal rhythm of plants is primarily driven by UV. The change from -UV conditions to +UV conditions leads to a relatively rapid increase in flavonoid content over several days. In contrast, the flavonoid content does not drop as quickly when changing to -UV conditions and initially remains at a steady-state level before slowly decreasing. Plants can start the diurnal rhythm at the beginning of the day (9:00 CDT), when the soluble flavonoid glycosides and hydroxycinnamic acid derivatives are still present in low concentrations, regardless of whether the plants were previously adapted to UV or not. Later in the day (13:00 CDT), such rapid adaptation is no longer possible and plants can only start with the diurnal rhythm the following day. The transfer of these results to other crop plants needs to be investigated in the future.

## Figures and Tables

**Figure 1 plants-10-02268-f001:**
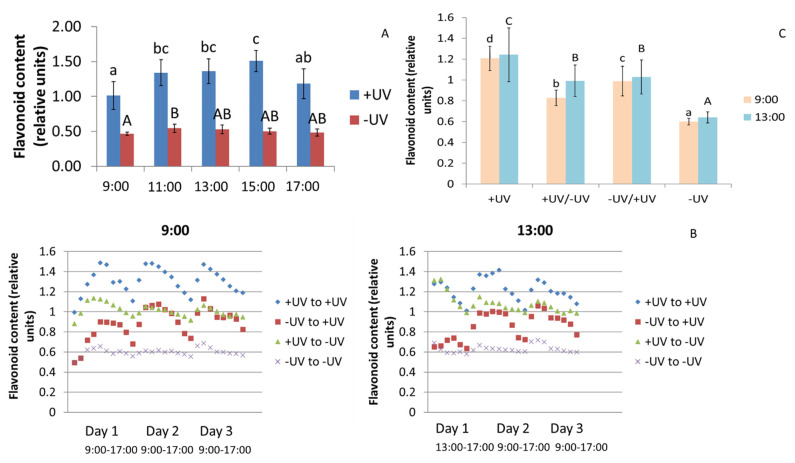
Flavonoid contents measured optically (Dualex) in okra plants (**A**) in a one-day diurnal rhythm (Experiment 1—targeting diurnal rhythm) and (**B**) a follow-up for three days after a transfer of okra plants at 9:00 and 13:00 (Experiment 2—targeting changes in diurnal rhythm due to transfer to the opposite UV condition) and (**C**) after an adaptation for seven days at 9:00 and 13:00 (Experiment 2—targeting changes in diurnal rhythm due to transfer to the opposite UV condition). Different letters represent significant differences at *p* ≤ 0.05.

**Figure 2 plants-10-02268-f002:**
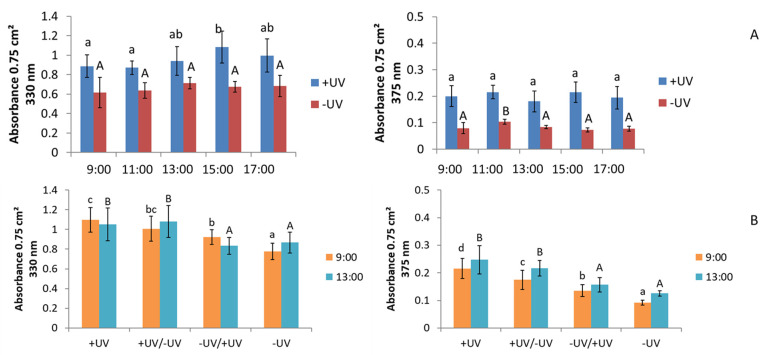
Absorbance of okra plant extracts at 330 nm and 375 nm dependent on (**A**) the diurnal rhythm (Experiment 1–targeting diurnal rhythm) and (**B**) the UV conditions at transfer and harvest times of 9:00 and 13:00 with a 7-day adaption period after the transfer (Experiment 2–targeting changes in diurnal rhythm due to transfer to the opposite UV condition). Different letters represent significant differences at *p* ≤ 0.05.

**Figure 3 plants-10-02268-f003:**
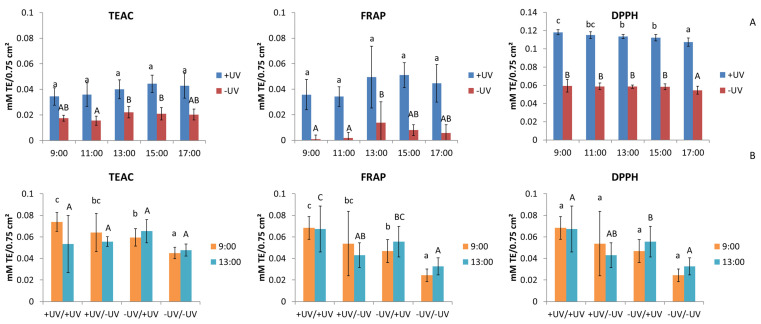
Antioxidant activity of okra plant extracts with TEAC, FRAP, or DPPH assay dependent on (**A**) the diurnal rhythm (Experiment 1—targeting diurnal rhythm) and (**B**) the UV conditions at transfer and harvest times of 9:00 and 13:00 with a 7-day adaption period after the transfer (Experiment 2—targeting changes in diurnal rhythm due to transfer to the opposite UV condition). Different letters represent significant differences at *p* ≤ 0.05.

**Table 1 plants-10-02268-t001:** Morphological parameters of okra plants grown under different UV conditions in Experiment 1 (targeting diurnal rhythm) and Experiment 2 (targeting changes in diurnal rhythm due to transfer to the opposite UV condition). These parameters were presented at 13:00 CDT (afternoon) for Experiment 1, and at 9:00 CDT (morning) and 13:00 CDT (afternoon) for Experiment 2. Different letters represent significant differences between the UV conditions at *p* ≤ 0.05.

	Experiment 1		Experiment 2							
	13:00		9:00				13:00			
	+UV	-UV	+UV	+UV/-UV	-UV/+UV	-UV	+UV	+UV/-UV	-UV/+UV	-UV
Plant height (cm)	23.14 ± 3.86 a	27.07 ± 1.29 b	20.40 ± 1.14	20.73 ± 8.82	20.08 ± 1.84	21.29 ± 1.08	19.46 ± 2.42	19.47 ± 2.53	20.93 ± 1.82	21.38 ± 1.27
Internode length (cm)	4.53 ± 1.34 a	5.88 ± 0.65 b	2.45 ± 0.38 A	3.0 ± 0.76 AB	2.71 ± 0.68 AB	3.21 ± 1.05 B	2.41 ± 0.35	2.49 ± 0.25	2.60 ± 0.29	3.02 ± 0.62
Leaf area (cm^2^)	61.76 ± 9.91	63.50 ± 10.25	36.25 ± 6.36	38.10 ± 6.70	33.98 ± 5.40	45.25 ± 5.57	37.83 ± 8.73	37.28 ± 6.27	43.40 ± 6.61	44.57 ± 8.48
Fresh matter (g/leaf)	1.27 ± 0.33	1.40 ± 0.17	0.94 ± 0.17 AB	0.99 ± 0.18 AB	0.86 ± 0.13 A	1.09 ± 0.14 B	0.95 ± 0.23	0.93 ± 0.14	1.05 ± 0.16	1.09 ± 0.20
Dry matter (g/leaf)	0.22 ± 0.05	0.27 ± 0.03	0.14 ± 0.02 AB	0.15 ± 0.02 AB	0.13 ± 0.02 A	0.17 ± 0.02 B	0.16 ± 0.03	0.15 ± 0.03	0.17 ± 0.03	0.18 ± 0.04
Dry matter (mg/0.75 cm^2^)	2.18 ± 0.35	2.26 ± 0.24	2.26 ± 0.42	2.22 ± 0.28	2.33 ± 0.24	2.09 ± 0.34	2.47 ± 0.34	2.21 ± 0.40	2.25 ± 0.22	2.36 ± 0.23

**Table 2 plants-10-02268-t002:** Soluble flavonoid glycosides and hydroxycinnamic acids in okra grown under different UV conditions at different times of the day in Experiment 1 (targeting diurnal rhythm). Different letters represent significant differences between the times of a day at *p* ≤ 0.05. I: isorhamnetin; K: kaempferol; Q: quercetin; glc: glucose; xyl: xylose; HFer: hydroxyferuloly; Sin: sinapoyl.

Experiment 1										
	9:00		11:00		13:00		15:00		17:00	
	+UV	-UV	+UV	-UV	+UV	-UV	+UV	-UV	+UV	-UV
HFer-glc	4.78 ± 1.95 bc	2.48 ± 1.42 B	6.45 ± 1.50 c	6.25 ± 0.60 C	2.45 ± 1.75 a	0.75 ± 0.58 A	3.54 ± 2.33 ab	2.14 ± 1.26 B	6.36 ± 1.24 c	5.47 ± 0.97 C
I-3-glc	0.03 ± 0.03 a	0.00 ± 0.01 A	0.09 ± 0.02 c	0.02 ± 0.01 B	0.01 ± 0.01 a	0.00 ± 0.00 A	0.01 ± 0.01 a	0.00 ± 0.00 A	0.06 ± 0.02 b	0.01 ± 0.01 A
K-3-glc	0.15 ± 0.07 a	0.07 ± 0.13 A	0.28 ± 0.07 b	0.08 ± 0.03 A	0.07 ± 0.05 a	0.01 ± 0.01 A	0.11 ± 0.08 a	0.02 ± 0.01 A	0.28 ± 0.12 b	0.08 ± 0.04 A
K-3-glc-xyl	0.38 ± 0.22 b	0.09 ± 0.07 A	0.74 ± 0.12 c	0.26 ± 0.09 B	0.08 ± 0.06 a	0.02 ± 0.01 A	0.18 ± 0.14 a	0.04 ± 0.03 A	0.54 ± 0.14 b	0.18 ± 0.09 B
Q-3-glc	0.28 ± 0.18 ab	0.01 ± 0.01 A	0.63 ± 0.14 c	0.02 ± 0.01 B	0.08 ± 0.07 a	0.00 ± 0.00 A	0.19 ± 0.15 a	0.00 ± 0.00 A	0.48 ± 0.23 bc	0.02 ± 0.02 A
Q-3-glc-xyl	3.90 ± 2.67 ab	0.15 ± 0.14 A	8.56 ± 3.22 c	0.44 ± 0.20 B	0.97 ± 1.16 a	0.03 ± 0.02 A	2.04 ± 1.85 a	0.08 ± 0.06 A	6.02 ± 2.50 bc	0.35 ± 0.22 B
Sin-glc	0.57 ± 0.32 a	0.11 ± 0.02 A	1.10 ± 0.49 b	0.18 ± 0.02 A	0.43 ± 0.18 a	0.13 ± 0.03 A	0.35 ± 0.09 a	0.19 ± 0.18 A	0.55 ± 0.36 a	0.15 ± 0.03 A

**Table 3 plants-10-02268-t003:** Soluble flavonoid glycosides and hydroxycinnamic acids in okra grown under different UV conditions on day seven at different times of the day in Experiment 2 (targeting changes in diurnal rhythm due to transfer to the opposite UV condition). Different letters represent significant differences between the UV conditions at *p* ≤ 0.05. I: isorhamnetin; K: kaempferol; Q: quercetin; glc: glucose; xyl: xylose; HFer: hydroxyferuloly; Sin: sinapoyl.

Experiment 2								
	9:00				13:00			
	+UV	+UV/-UV	-UV/+UV	-UV	+UV	+UV/-UV	-UV/+UV	-UV
HFer-glc	6.44 ± 0.71	6.56 ± 0.24	6.69 ± 0.40	6.31 ± 0.75	6.42 ± 0.86	6.18 ± 0.44	6.66 ± 0.73	6.16 ± 0.31
I-3-glc	0.05 ± 0.02 b	0.03 ± 0.01 a	0.03 ± 0.01 a	0.07 ± 0.04 b	0.08 ± 0.04 B	0.01 ± 0.01 A	0.05 ± 0.03 B	0.01 ± 0.01 A
K-3-glc	0.13 ± 0.07	0.12 ± 0.05	0.14 ± 0.09	0.13 ± 0.07	0.16 ± 0.04 B	0.07 ± 0.02 A	0.14 ± 0.04 B	0.05 ± 0.02 A
K-3-glc-xyl	0.49 ± 0.23	0.46 ± 0.14	0.48 ± 0.14	0.42 ± 0.23	0.59 ± 0.20 B	0.34 ± 0.08 A	0.58 ± 0.15 B	0.29 ± 0.05 A
Q-3-glc	0.23 ± 0.19 a	0.27 ± 0.09 ab	0.25 ± 0.11 ab	0.43 ± 0.24 b	0.35 ± 0.12 C	0.15 ± 0.09 B	0.28 ± 0.13 C	0.02 ± 0.01 A
Q-3-glc-xyl	6.61 ± 2.05	4.70 ± 1.28	5.28 ± 1.89	7.15 ± 3.28	7.61 ± 1.92 C	2.92 ± 0.91 B	6.23 ± 1.98 C	0.55 ± 0.16 A
Sin-glc	1.59 ± 0.08 c	1.37 ± 0.11 ab	1.45 ± 0.17 bc	1.18 ± 0.27 a	1.48 ± 0.16 B	1.16 ± 0.25 B	1.35 ± 0.27 B	0.13 ± 0.40 A

**Table 4 plants-10-02268-t004:** Correlation coefficients of UV absorption at 330 and 370 nm in absorbance per 0.75 cm^2^ and antioxidant activity (TEAC, FRAP, and DPPH) in mM Troxolequivalent per g dry matter, soluble flavonoid glycosides, and hydroxycinnamic acids in mg per g dry matter, and flavonoid content (FLAV) in relative arbitrary units (Dualex) in +UV plants from Experiment 1 (targeting diurnal rhythm). Bold values represent significant correlations at *p* ≤ 0.05. I: isorhamnetin; K: kaempferol; Q: quercetin; glc: glucose; xyl: xylose; HFer: hydroxyferuloly; Sin: sinapoyl.

+UV	330	370	TEAC	FRAP	DPPH	HFer-glc	I-3-glc	K-3-glc	K-3-glc-xyl	Q-3-glc	Q-3-glc-xyl	Sin-glc	FLAV
330	1.00	**0.66**	**0.71**	**0.73**	**−0.36**	**−0.30**	−0.19	0.01	−0.27	−0.07	−0.24	**−0.30**	0.19
370		1.00	**0.63**	**0.54**	−0.08	−0.02	**0.30**	**0.31**	0.23	**0.38**	**0.31**	0.18	0.10
TEAC			1.00	**0.75**	**−0.34**	−0.13	0.07	0.10	−0.07	0.11	0.06	0.00	0.26
FRAP				1.00	**−0.34**	**−0.29**	−0.14	−0.11	−0.27	−0.10	−0.14	−0.20	0.23
DPPH					1.00	−0.07	−0.02	−0.08	0.04	−0.02	−0.02	0.19	−0.21
HFer-glc						1.00	**0.64**	**0.67**	**0.74**	**0.68**	**0.66**	**0.28**	−0.15
I-3-glc							1.00	**0.84**	**0.93**	**0.92**	**0.91**	**0.57**	0.00
K-3-glc								1.00	**0.87**	**0.91**	**0.78**	**0.37**	−0.11
K-3-glc-xyl									1.00	**0.90**	**0.90**	**0.59**	−0.17
Q-3-glc										1.00	**0.90**	**0.41**	−0.10
Q-3-glc-xyl											1.00	**0.61**	−0.04
Sin-glc												1.00	−0.04
FLAV													1.00

**Table 5 plants-10-02268-t005:** Correlation coefficients of UV-absorption at 330 and 370 nm in absorbance per 0.75 cm^2^ and antioxidant activity (TEAC, FRAP, and DPPH) in mM Troxolequivalent per g dry matter, soluble flavonoid glycosides and hydroxycinnamic acids in mg per g dry matter, and flavonoid content (FLAV) in relative arbitrary units (Dualex) in -UV plants from Experiment 1 (targeting diurnal rhythm). Bold values represent significant correlations at *p* ≤ 0.05. I: isorhamnetin; K: kaempferol; Q: quercetin; glc: glucose; xyl: xylose; HFer: hydroxyferuloly; Sin: sinapoyl.

-UV	330	370	TEAC	FRAP	DPPH	HFer-glc	I-3-glc	K-3-glc	K-3-glc-xyl	Q-3-glc	Q-3-glc-xyl	Sin-glc	FLAV
330	1.00	**0.51**	**0.51**	**0.46**	−0.11	−0.07	0.01	−0.08	−0.08	−0.01	−0.07	0.01	0.24
370		1.00	−0.16	0.05	0.12	**0.38**	**0.39**	0.09	**0.41**	**0.31**	0.34	0.06	**0.38**
TEAC			1.00	**0.63**	−0.16	−0.26	−0.17	−0.12	−0.20	−0.22	−0.14	0.05	0.12
FRAP				1.00	−0.01	−0.24	−0.21	−0.17	−0.17	−0.14	−0.16	−0.04	**0.43**
DPPH					1.00	−0.28	−0.05	−0.21	−0.16	**−0.28**	−0.23	−0.16	0.16
HFer-glc						1.00	**0.54**	**0.44**	**0.91**	**0.78**	**0.87**	0.10	0.10
I-3-glc							1.00	**0.20**	**0.51**	**0.36**	**0.49**	0.12	0.23
K-3-glc								1.00	**0.49**	**0.39**	**0.50**	0.03	−0.03
K-3-glc-xyl									1.00	**0.78**	**0.97**	0.07	0.17
Q-3-glc										1.00	**0.79**	0.08	0.16
Q-3-glc-xyl											1.00	0.05	0.18
Sin-glc												1.00	−0.12
FLAV													1.00

## Data Availability

The data presented in this study are openly available at https://data.goettingen-research-online.de/dataset.xhtml?persistentId=doi:10.25625/VKZAME (accessed on 15 September 2021).

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
