# Peer review of "The Function of Flavonoids in the Diurnal Rhythm under Rapidly Changing UV Conditions—A Model Study on Okra"

_plants, 2021, doi:10.3390/plants10112268_

Round 1
Reviewer 1 Report
The manuscript provides the results of a study that evaluated varying UV conditions on the flavonoid levels and oxidation potential of okra. The main finding was that flavonoid levels and their associated functions are subject to changes that appear to be regulated in some ways by the circadian clock. That information in and of itself warrants publication. However, the manuscript tends to ramble, not allowing the reader a chance to fully appreciate the data present.
The manuscript is publishable but in need of major revision. Key suggestions are as follows:
1. Figure 1 needs to contain a graphical depiction of the experiments performed and the measurements taken. Such a figure will help frame Experiments 1 and 2 early on. Also note the current Fig 1B has no x-axes listed and the units for flavonoid content are not provided. The content appears to be relative since it relies upon the fluorescence of a surface.
2. The first paragraph of the introduction is essentially a microcosm of the problem with the paper. It rambles from point to point without a consistent through line. Is the paragraph even necessary? The second paragraph would serve nicely as a starting point, with reference to environmental changes.
3. The tables are unapproachable and the reader is left to review the text, which tends to be somewhat non-descriptive. Some have soluble flavonoid glycosides listed, but one is unsure of what the number means.
Author Response
Thank you for your comments
The manuscript provides the results of a study that evaluated varying UV conditions on the flavonoid levels and oxidation potential of okra. The main finding was that flavonoid levels and their associated functions are subject to changes that appear to be regulated in some ways by the circadian clock. That information in and of itself warrants publication. However, the manuscript tends to ramble, not allowing the reader a chance to fully appreciate the data present.
The manuscript is publishable but in need of major revision. Key suggestions are as follows:
- Figure 1 needs to contain a graphical depiction of the experiments performed and the measurements taken. Such a figure will help frame Experiments 1 and 2 early on. Also note the current Fig 1B has no x-axes listed and the units for flavonoid content are not provided. The content appears to be relative since it relies upon the fluorescence of a surface.
We now do provide a graphical visualization of the experimental set up. Nevertheless, we decided to do this as supplemental figure as there is already a high number of figures and tables in the manuscript. We also followed the suggestion of another reviewer to specify the experiments purpose in the figure’s and table’ description.
- The first paragraph of the introduction is essentially a microcosm of the problem with the paper. It rambles from point to point without a consistent through line. Is the paragraph even necessary? The second paragraph would serve nicely as a starting point, with reference to environmental changes.
This paragraph has been tightened in light of climate change and the UV conditions that accompany it. We think that this paragraph is necessary to explain the outdoor experiment instead of experiments under defined conditions.
- The tables are unapproachable and the reader is left to review the text, which tends to be somewhat non-descriptive. Some have soluble flavonoid glycosides listed, but one is unsure of what the number means.
We now included more information in the figure’s and table’ description. We hope this also helps solving the issue with the numbers and abbreviations.
Reviewer 2 Report
The manuscript presents interesting investigations about plants'responses to UV exposure. However, there are some minor changes that should be made, as presented below.
- Please make it clear what you mean by "+UV" and "-UV". Are you referring to plants grown in the presence and absence of UV radiation? Or is it related to the intensity of the UV radiation?
- What type of UV radiation passes through the plastic films used (UV-A, UV-B, or both)? More characteristics about the plastic films should be given, if known.
- In Figure 2, I suggest changing the scale for the absorbance at 375 nm. I do not see the point of using a scale of up to 1 or 1.2 when the values stop around 0.3. A scale up to 0.4 or 0.5 should be more than enough, and the difference between the represented values would be more visibil.
Author Response
Thank you for your comments.
The manuscript presents interesting investigations about plants'responses to UV exposure. However, there are some minor changes that should be made, as presented below.
- Please make it clear what you mean by "+UV" and "-UV". Are you referring to plants grown in the presence and absence of UV radiation? Or is it related to the intensity of the UV radiation?
Thank you for pointing out it was missing. It is now clarified in the material and methods part.
- What type of UV radiation passes through the plastic films used (UV-A, UV-B, or both)? More characteristics about the plastic films should be given, if known.
We refer to Neugart et al. 2021 in which the spectra of the films are given.
- In Figure 2, I suggest changing the scale for the absorbance at 375 nm. I do not see the point of using a scale of up to 1 or 1.2 when the values stop around 0.3. A scale up to 0.4 or 0.5 should be more than enough, and the difference between the represented values would be more visibil.
Done
Reviewer 3 Report
The manuscript deals with the responses of the Okra (Abelmoschus esculentus) plant to UV light from the point of view of the formation of flavonoids as UV absorbing substances and changes in the diurnal rhythm of the plants. It is obvious that the research was carried out carefully and originally in natural conditions, but the presentation of the research in the manuscript could have been much better.
My comments:
- The title sounds too general, it does not refer at all to the object on which the research was done, also the abstract could be written more clearly and systematically.
- The introduction is very chaotic, the authors did not present at all the okra plant on which the research was done, the reason for its selection for the research conducted, the uniqueness of its metabolites.
- The publication contains a lot of tables with a lot of numerical data, it is difficult to navigate through them.
- The authors conducted research under field conditions, but today there are many growing facilities with well-defined conditions available, from light spectrum, to temperature control, humidity, etc... To use natural conditions with UV filters is certainly interesting, but would be more suited to agronomy-focused journals.
- The conclusions could have been formulated better, not everything was clear from the text. In fact, the most interesting result from my point of view is on lines 220 to 221, that "plants transferred at midday do not begin an altered diurnal rhythm until the next day. " Of course, other experts will come to different parts of the results.
The results could also be supported by analysis of transcription of key genes for flavonoid biosynthesis.
Experiment 1 and experiment 2 seem somewhat inconsistent. Perhaps both experiments should be re-titled according to their purpose.
line 49 - typo: favonol
line 380 – typo: dirunal
Much is already known about the importance of UV conditions on food quality, but is it really significant under purely natural conditions? Does this refer to medicinal plants or also to the nutritional quality of the fruits etc.?
The two graphs of the Figure 1C could be combined together.
Couldn't abbreviations like TEAC, FRAP, DPPH and many others be explained in a clear way?
Finally, I'm not sure this article fits well to the Plants journal. If the authors make it more clear and systematic as well as more readable, it could be published.
Author Response
Thank you for your comments.
The manuscript deals with the responses of the Okra (Abelmoschus esculentus) plant to UV light from the point of view of the formation of flavonoids as UV absorbing substances and changes in the diurnal rhythm of the plants. It is obvious that the research was carried out carefully and originally in natural conditions, but the presentation of the research in the manuscript could have been much better.
My comments:
- The title sounds too general, it does not refer at all to the object on which the research was done, also the abstract could be written more clearly and systematically.
Both were reworded.
- The introduction is very chaotic, the authors did not present at all the okra plant on which the research was done, the reason for its selection for the research conducted, the uniqueness of its metabolites.
The introduction is sharpened now and we included a paragraph on okra.
- The publication contains a lot of tables with a lot of numerical data, it is difficult to navigate through them.
We now have more information given in the figure’s and table’ description. We hope that this supports the understanding.
- The authors conducted research under field conditions, but today there are many growing facilities with well-defined conditions available, from light spectrum, to temperature control, humidity, etc... To use natural conditions with UV filters is certainly interesting, but would be more suited to agronomy-focused journals.
Since there is lots of studies on artificial light sources there is also the need to study these effects under natural condition. We definitely see the benefit of well-defined conditions. However, a lot of studies work with unrealistic UV:PAR ratios as their light sources do not provide a combined use. New LED modules now offer the chance to overcome this problem. However, UV-B LEDs are very expensive and often companies do not yet have much experience with mounting UV-B LEDs to chips. Special materials are needed for artificial light sources including UV-B LEDs. On top of that the intensities are still not as high as in sunlight. So from the light perspective the sunlight offers the natural ratios and we think such studies are needed as well to investigate the extent to which plants can react to UV. As you see from the data the temperature effect is only very small with maximum changes of 5°C throughout the day.
- The conclusions could have been formulated better, not everything was clear from the text. In fact, the most interesting result from my point of view is on lines 220 to 221, that "plants transferred at midday do not begin an altered diurnal rhythm until the next day. " Of course, other experts will come to different parts of the results.
This outcome is highlighted now in the conclusion due to rewriting and reordering of the text.
The results could also be supported by analysis of transcription of key genes for flavonoid biosynthesis.
We totally agree to this statement. We also discussed that during the study design. Nevertheless, several studies already stated that measurements on gene expression and metabolites should be done at different time point due to their interlink. Which means we would have had different leaves or even plants for this measurement to avoid wounding stress since both methods are destructive. The outstanding characteristic of this study was to have the optical measurement and the destructive measurements in the exact same spot of the leaf in experiment 1. Therefore, we worked with leaf discs which is not a lot material for the measurements done. In experiment 2 we did the optical measurements on the exact same spot of the leaf in a follow-up procedure. Here we also used the spot of optical measurement for the destructive measurements.
Experiment 1 and experiment 2 seem somewhat inconsistent. Perhaps both experiments should be re-titled according to their purpose.
We now state the purpose of the experiment in the text after naming the experiment.
line 49 - typo: favonol
line 380 – typo: diurnal
thank you. The mistakes were corrected and the manuscript checked.
Much is already known about the importance of UV conditions on food quality, but is it really significant under purely natural conditions? Does this refer to medicinal plants or also to the nutritional quality of the fruits etc.?
There is a number of plants showing a diurnal rhythms. How and to which extent they do this is not yet clear. We highlighted more that this is a model study on okra, which is a crop plant as well. It is very likely that the gained knowledge can be transferred to other plants, especially leafy vegetables, as orka has very common quercetin and kaempferol glycosides as well as hydroxycinnamic acids. Nevertheless, we do not know if fruits follow this diurnal rhythm as well.
The two graphs of the Figure 1C could be combined together.
Done
Couldn't abbreviations like TEAC, FRAP, DPPH and many others be explained in a clear way?
The abbreviations are explained now.
Finally, I'm not sure this article fits well to the Plants journal. If the authors make it more clear and systematic as well as more readable, it could be published.
This manuscript was submitted the Special Issue "The Multifaceted Responses of Plants to Visible and Ultraviolet Radiation". We think it highlights the plant-environment interactions and also targets climate change aspects. To our best knowledge the manuscript fits into the Scope.
Round 2
Reviewer 1 Report
The authors addressed my comments. However, the Supplemental Figure describing the Expts is not mentioned early in the paper (besides the Abstract). Thus one has no clue what each experiment is. The authors need to have a paragraph describing Expt 1 and 2 at the end of the Introduction or the beginning of the results section and refer to the somewhat confusing Supplemental Figure (one could do a better job with that). Once that is done, I am okay with the paper.
Author Response
The authors addressed my comments. However, the Supplemental Figure describing the Expts is not mentioned early in the paper (besides the Abstract). Thus one has no clue what each experiment is. The authors need to have a paragraph describing Expt 1 and 2 at the end of the Introduction or the beginning of the results section and refer to the somewhat confusing Supplemental Figure (one could do a better job with that). Once that is done, I am okay with the paper.
The figure was rearranged and we now included an introductory part in the results section.
Reviewer 3 Report
I am pleased to see that the manuscript has been improved considerably. I appreciate the addition of information on the okra plant, but it is still not entirely clear to me what has made this species so popular in research on UV responses. In general, I would justify a little more the importance of flavonoids to the pharmaceutical industry, which would increase the importance of the research presented in this manuscript. Regarding the aims of the paper, I was surprised that the authors based the change in flavonoid content as the primary factor and not the UV radiation itself. In the supplements, it would be nice to see the spectral light curves measured by somespectroradiometer in areas without and under filters to get an idea of how efficient the filters were
Typos: line 99 - were investigated instead of was
line 104 - thaat
Fig. 1 - the graphs could always be rearranged two side by side
line 437 - omit the word excelently and replace with reliably
Author Response
I am pleased to see that the manuscript has been improved considerably. I appreciate the addition of information on the okra plant, but it is still not entirely clear to me what has made this species so popular in research on UV responses.
We added two sentences to underline why okra was choosen.
In general, I would justify a little more the importance of flavonoids to the pharmaceutical industry, which would increase the importance of the research presented in this manuscript.
We added the importance of flavonoids and their functions in the human body in the flavonoid paragraph.
Regarding the aims of the paper, I was surprised that the authors based the change in flavonoid content as the primary factor and not the UV radiation itself.
It is well known that changing UV conditions can change the concentration of flavonoids. However, it is seldom discussed what the consequences of these changes are and how the functions of flavonoids in the plant are affected by these changes. That is why we choose to focus the aims this way. Nevertheless, it is possible for the reader to draw conclusions based on the UV-condition.
In the supplements, it would be nice to see the spectral light curves measured by somespectroradiometer in areas without and under filters to get an idea of how efficient the filters were
A supplemental figure was added.
Typos: line 99 - were investigated instead of was
Thank you
line 104 - that
Thank you
Fig. 1 - the graphs could always be rearranged two side by side
Done
line 437 - omit the word excelently and replace with reliably
Done